# vMF Guided learning for biomedical Vision-Language Models

## Abstract

Effective adaptation of Vision–Language Models (VLMs) to biomedical tasks remains challenging due to a substantial semantic gap between general knowledge and domain-specific expertise. Domain-specific models such as BiomedCLIP narrow this gap; however, prevailing prompt-learning methods collapse diverse text embeddings into a single prototype, discarding distributional information. We introduce vMF Distribution Semantic Alignment (VDSA), which models each class with a von Mises–Fisher distribution on the unit hypersphere and aligns images to the entire distribution rather than a single prototype. We further derive a closed-form upper bound to the expected contrastive loss, yielding a sampling-free objective that is implicitly equivalent to aligning against an infinite prompt ensemble with minimal overhead. Experiments on multiple biomedical benchmarks show that VDSA consistently improves few-shot adaptation and generalization to unseen classes, providing a robust recipe for adapting specialized VLMs.

## 1 Introduction

Vision–Language Models (VLMs), such as CLIP (Radford et al., 2021) and Align (Jia et al., 2021), have established a powerful paradigm for learning unified image–text representations. By leveraging large-scale image–text training, they achieve robust semantic alignment and impressive zero-shot transfer, motivating extensive research on their adaptation to diverse downstream tasks (Zhang et al., 2021; Zhou et al., 2022b;a). However, transferring this success to highly specialized domains such as biomedicine reveals a fundamental challenge: a substantial semantic gap between general world knowledge and domain-specific biomedical expertise (Zhang et al., 2023; Eslami et al., 2021). This gap spans both visual and textual modalities. Unlike natural images, the visual language of biomedical imaging (e.g., X-rays, MRIs) is abstract and non-intuitive, characterized not by common semantic categories but by subtle, intricate patterns of texture, intensity, and anatomical structure that require specialized training to interpret (Koleilat et al., 2025). As a result, the generic visual concepts captured by standard VLMs often fail to align with these domain-specific patterns. Similarly, the medical terminology used to describe such findings (e.g., "pleural effusion," "nodular sclerosis") lies far beyond the common web-scale vocabulary, thereby limiting the applicability of general-purpose text encoders (Zhang et al., 2023).

To bridge the semantic gap in biomedicine, models pre-trained on domain-specific data are indispensable. Foundation models such as BiomedCLIP (Zhang et al., 2023), trained on 15 million biomedical image–text pairs, provide a far stronger starting point for downstream medical tasks than general-purpose VLMs. However, adapting these models through full fine-tuning is often computationally prohibitive and prone to overfitting or catastrophic forgetting (Gao et al., 2024). This challenge motivates the use of parameter-efficient adaptation strategies that update only a small subset of parameters while preserving the knowledge captured during pretraining (Zhou et al., 2022b; Bafghi et al., 2025).

Beyond the choice of adaptation mechanism, the effectiveness of VLMs depends critically on the design of text prompts. Early approaches rely on a single handcrafted prompt (e.g., "a photo of a [CLASS]"), which is sensitive to phrasing and limited in expressiveness (Radford et al., 2021). Prompt ensembling alleviates this by combining multiple templates to improve robustness (Kim et al., 2025), while recent advances leverage large language models (LLMs) to automatically generate diverse, domain-specific prompts (Pratt et al., 2023). These strategies enrich the semantic space

available to the model and have proven to benefit prompt learning by bringing diverse text representations (Zheng et al., 2024). Alternatively, approaches such as CoOp (Zhou et al., 2022b) and CoCoOp (Zhou et al., 2022a) learn continuous prompt embeddings directly by treating the prompt tokens as trainable parameters.

Despite these advances, current prompt ensembling methods share a fundamental limitation in how they aggregate semantic diversity. A finite set of $N$ prompt embeddings is ultimately collapsed into a single prototype, a lossy compression that preserves only the semantic center while discarding crucial information about the dispersion and angular structure of the distribution (Radford et al., 2021; Allingham et al., 2023). This restricts the model to align with a narrow point estimate of class semantics rather than the full semantic space. Moreover, the choice of $N$ presents a costly trade-off between semantic coverage and efficiency, with no principled way to determine an optimal value.

We attribute this limitation to representing semantic diversity as a finite set of discrete prompt embeddings that are merely averaged. A more principled alternative is to model the underlying continuous probability distribution that the discrete prompt set approximates (Ma et al., 2023). By shifting the alignment objective from a single prototype to the full distribution, the model can capture not only the semantic center but also the dispersion and angular structure that encode diversity and ambiguity. This perspective effectively enables the model to learn from the complete semantic space, as if integrating over an unlimited ensemble of text embeddings, while avoiding the prohibitive cost of explicitly sampling a large number of prompts.

To realize distributional alignment, we propose **vMF Distribution Semantic Alignment (VDSA)**, which interprets a class's $N$ prompts as samples from a latent semantic distribution on the unit hypersphere and estimates its parameters by maximum likelihood. Instead of aligning image features to a single prototype, VDSA aligns them to the entire class distribution. The exact expected contrastive loss is analytically intractable because it involves high-dimensional surface integrals on the hypersphere induced by class-specific vMF distributions (Mardia & Jupp, 2009) for which no closed form exists. To address this challenge, we derive a closed-form Jensen upper bound that is fully differentiable and free of sampling, which encourages embeddings to the high-density regions of the semantic space. For computational efficiency, only a small subset of vision encoder weights is updated through LoRA (Hu et al., 2022). Conceptually, VDSA can be viewed as aligning against an infinite ensemble of prompts, providing implicit semantic augmentation that surpasses discrete prompt ensembling. Evaluations across diverse biomedical benchmarks demonstrate that VDSA achieves state-of-the-art performance in both few-shot adaptation and base-to-novel generalization.

## 2 RELATED WORK

### 2.1 VISION–LANGUAGE MODELS

Vision–Language Models (VLMs) such as CLIP (Radford et al., 2021) and ALIGN (Jia et al., 2021) have shown that large-scale image–text pretraining yields robust representations with strong zero-shot generalization. Recent extensions in the biomedical domain include MedCLIP (Wang et al., 2022), PubMedCLIP (Eslami et al., 2021), and BiomedCLIP (Zhang et al., 2023). BioMed-CLIP (Zhang et al., 2023) incorporates millions of domain-specific image–text pairs, achieving notable gains over general-purpose VLMs on medical benchmarks. Despite these advances, their utility on specialized biomedical tasks remains limited, as capturing subtle, disease-specific semantics often requires additional adaptation beyond pretraining. This motivates the development of methods that can more effectively tailor biomedical foundation models to the demands of target clinical applications (Koleilat et al., 2025).

### 2.2 PARAMETER-EFFICIENT ADAPTATION

Adapting large-scale foundation models to downstream tasks through full fine-tuning is often computationally prohibitive and risks overfitting or catastrophic forgetting (Ding et al., 2022). To address this, a range of parameter-efficient adaptation (PEFT) strategies has been proposed. Adapter-based methods (Gao et al., 2024; Zhang et al., 2021) introduce lightweight modules between transformer layers, while low-rank adaptation (LoRA) (Hu et al., 2022) injects trainable rank-decomposed matrices into weight updates. Linear probing simply freezes the pretrained encoder and trains only a linear classifier on top of the fixed feature representation (Huang et al., 2024b). These techniques

preserve most pretrained weights, enabling efficient transfer across diverse domains. In biomedical vision–language tasks, PEFT has also shown promise in tailoring domain-specific foundation models with limited supervision (Peng et al., 2025). However, while PEFT mitigates the cost of adaptation, it primarily operates on the model parameters themselves and does not directly address how semantic prompts are constructed. Since the quality of prompts critically determines the alignment between image and text, optimizing prompt design remains an orthogonal yet equally crucial direction.

## 2.3 Prompt Ensembling and Learning

The design of text prompts plays a decisive role in the effectiveness of vision–language models. Early approaches relied on manually crafted templates, which are simple but sensitive to phrasing and lack semantic richness (Radford et al., 2021). To improve robustness, prompt ensembling has been widely adopted (Allingham et al., 2023; Roth et al., 2023), where multiple templates are used to provide diverse textual descriptions. Recent work further leverages large language models (LLMs) to automatically generate domain-specific prompt ensembles (Pratt et al., 2023; Koleilat et al., 2025), alleviating the need for manual design.

In parallel, learnable prompt methods such as CoOp (Zhou et al., 2022b) and CoCoOp (Zhou et al., 2022a) replace fixed templates with continuous embeddings that are optimized end-to-end, enabling more flexible adaptation to downstream tasks. Building on this direction, KgCoOp (Yao et al., 2023) and ProGrad (Zhu et al., 2023) further refine textual prompts by incorporating external knowledge and gradient-based guidance, respectively, to enhance model generalizability. Building on this line of work, BiomedCoOp (Koleilat et al., 2025) adapts prompt learning to biomedical images by combining a BiomedCLIP backbone with joint semantic–knowledge modeling through contextual mapping and selective prompt distillation, enabling more effective and robust domain-specific prompt context learning. Despite these advances, existing prompt ensembling and learnable prompt methods ultimately aggregate semantic information into a single prototype representation (Khattak et al., 2025; Huang et al., 2024a). Such compression inevitably discards distributional properties of the prompt set, including its dispersion and angular structure on the hypersphere. This limitation motivates approaches that move beyond discrete or mean-based representations toward principled distributional formulations of class semantics.

## 3 von Mises-Fisher Distribution Semantic Alignment

In this work, we introduce vMF Distribution Semantic Alignment (VDSA), a framework that advances contrastive language–image learning by enriching the representation of class semantics. VDSA generalizes beyond these formulations by modeling each class with a probability distribution over the hypersphere, thereby capturing its intrinsic semantic diversity. Specifically, we employ the von Mises–Fisher (vMF) distribution Mardia & Jupp (2009) to explicitly represent the semantic space of each class, and derive a closed-form upper bound of the expected contrastive loss under these distributions. This enables end-to-end optimization that implicitly achieves semantic augmentation and yields robust alignment.

### 3.1 Preliminaries

**Vision-Language Models**  CLIP (Radford et al., 2021) jointly trains an image encoder $\mathcal{E}_v$ and a text encoder $\mathcal{E}_t$ to learn a shared embedding space where paired images and texts are aligned. Given a batch of images, the encoder produces $\{\mathbf{z}_i\}_{i=1}^B$, with each $\mathbf{z}_i \in \mathbb{R}^D$. For a classification task with $C$ classes, a class-specific prompt template (e.g., "a photo of a {class_name}") is instantiated for each class and encoded into text features $\{\boldsymbol{\mu}_c\}_{c=1}^C$, where $\boldsymbol{\mu}_c \in \mathbb{R}^D$. All image and text features are $\ell_2$-normalized to lie on the unit hypersphere.

The alignment is enforced by a contrastive loss. For zero-shot classification, the probability of assigning image feature $\mathbf{z}_i$ to class $c$ is:

$$p(c|\mathbf{z}_i) = \frac{e^{s \cdot \mathbf{z}_i^\top \boldsymbol{\mu}_c}}{\sum_{j=1}^C e^{s \cdot \mathbf{z}_i^\top \boldsymbol{\mu}_j}}, \tag{1}$$

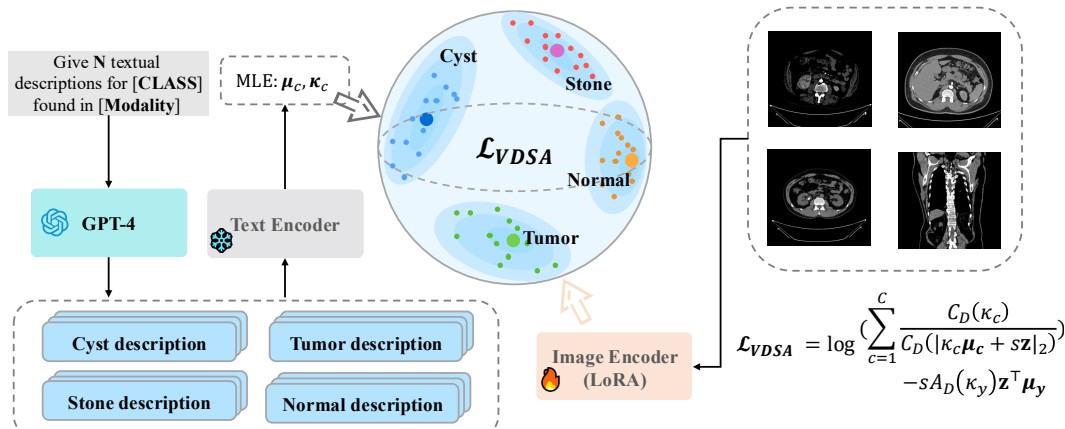

Figure 1: **Overview of our VDSA framework.** Our method models the semantics of each class as a von Mises–Fisher distribution on a hyperspherical feature space. We first leverage GPT-4 to generate $N$ diverse textual descriptions for each class, which are encoded into a set of feature vectors. These vectors are then used to fit a class-conditional vMF distribution $p(\mathbf{u} \mid \boldsymbol{\mu}_c, \kappa_c)$ via maximum likelihood estimation. An image encoder, fine-tuned with LoRA, is trained using our novel $\mathcal{L}_{\text{VDSA}}$ objective, which aligns image embeddings with their corresponding class distribution.

where $s > 0$ is a learnable temperature. The model is trained by minimizing the cross-entropy loss:

$$\mathcal{L}_{\text{CE}} = -\log p(y_i|\mathbf{z}_i). \tag{2}$$

In this formulation, each class $c$ is represented by a *single prototype vector* $\boldsymbol{\mu}_c$ on the hypersphere.

**Prompt Ensembling Method**   To address the limitations of a single prompt representation, *Prompt Ensembling* has been shown to benefit prompt learning by incorporating diverse textual descriptions for each class (Kim et al., 2025). By leveraging multiple prompts, ensembling provides a more robust and comprehensive semantic representation. Recent advances employ Large Language Models (LLMs) to automatically generate high-quality, domain-specific prompt ensembles, alleviating the need for manual template design (Khattak et al., 2025; Pratt et al., 2023). Following Biomed-CoOp (Koleilat et al., 2025), we adopt GPT-4 (Achiam et al., 2023) to synthesize class-specific prompts. For each dataset with $C$ classes, we query GPT-4 (Achiam et al., 2023) with the instruction:

> Give $N$ textual descriptions of visual discriminative features for distinct medical cases of [CLASS] found in [MODALITY].

This process yields $N$ diverse descriptions per class that capture characteristic lesions, anatomical patterns, and imaging cues, ensuring that the resulting prompts encode the necessary clinical semantics. For each class $c$, we generate $N$ diverse prompts, which are encoded by the text encoder $\mathcal{E}_t$ into a set of feature vectors $\{\boldsymbol{\mu}_c^{(i)}\}_{i=1}^N$, with each $\boldsymbol{\mu}_c^{(i)} \in \mathbb{R}^D$ lying on the unit hypersphere.

Following standard practice, these features are aggregated into a single prototype by computing their mean and subsequently re-normalizing to unit length:

$$\bar{\boldsymbol{\mu}}_c = \frac{\frac{1}{N}\sum_{i=1}^N \boldsymbol{\mu}_c^{(i)}}{\left\|\frac{1}{N}\sum_{i=1}^N \boldsymbol{\mu}_c^{(i)}\right\|_2}. \tag{3}$$

This ensemble prototype $\bar{\boldsymbol{\mu}}_c$ then replaces the single-prompt feature $\boldsymbol{\mu}_c$ in the contrastive loss (Eq. 1) for model adaptation.

As discussed earlier, the simple aggregation collapses $N$ prompt embeddings into prototypes; we instead model each class as a probability distribution on the unit hypersphere.

## 3.2 DISTRIBUTIONAL CLASS REPRESENTATION WITH vMF

To overcome the limitations of discrete prototypes, we lift the semantic representation of each class from a finite set of prompt embeddings to a continuous probability distribution, denoted $p(\mathbf{u} \mid \boldsymbol{\theta}_c)$, where $\mathbf{u} \in \mathcal{S}^{D-1}$ and $\boldsymbol{\theta}_c = (\boldsymbol{\mu}_c, \kappa_c)$. Here $\mathcal{S}^{D-1} = \{\mathbf{u} \in \mathbb{R}^D : \|\mathbf{u}\|_2 = 1\}$. This formulation provides a richer characterization of class semantics by capturing inherent diversity and ambiguity. Since CLIP features are $\ell_2$-normalized and thus lie on the unit hypersphere $\mathcal{S}^{D-1}$, the *von Mises–Fisher* (vMF) distribution is a natural choice, playing a role analogous to the Gaussian distribution in Euclidean space. See Appendix E for why Gaussian-based alternatives are unsuitable here.

The vMF distribution is parameterized by a mean direction $\boldsymbol{\mu}_c \in \mathcal{S}^{D-1}$ and a concentration parameter $\kappa_c \geq 0$. The mean direction $\boldsymbol{\mu}_c$ captures the semantic center of a class, while $\kappa_c$ controls its dispersion: larger $\kappa_c$ corresponds to a narrower, more coherent concept, whereas smaller $\kappa_c$ reflects broader variability.

Formally, the density for a random unit vector $\mathbf{u}$ is

$$p(\mathbf{u} \mid \boldsymbol{\mu}_c, \kappa_c) = C_D(\kappa_c)\, e^{\kappa_c\, \boldsymbol{\mu}_c^\top \mathbf{u}} \tag{4}$$

with normalization constant

$$C_D(\kappa_c) = \frac{\kappa_c^{\frac{D}{2}-1}}{(2\pi)^{\frac{D}{2}}\, I_{\frac{D}{2}-1}(\kappa_c)}, \tag{5}$$

where $I_\nu(\cdot)$ is the modified Bessel function of the first kind. Here $C_D(\kappa)$ is fixed by the normalization $\int_{\mathcal{S}^{D-1}} p(\mathbf{u} \mid \boldsymbol{\mu}, \kappa)\, d\sigma(\mathbf{u}) = 1$, where $d\sigma$ denotes the uniform surface-area measure on the sphere; evaluating this surface integral yields as shown in Equation 5. For efficiency, we compute $\log C_D(\cdot)$ via a numerically stable asymptotic approximation to $\log I_\nu(\cdot)$ (rather than explicit Bessel evaluations).

Given $N$ unit-normalized prompt embeddings $\{\boldsymbol{u}_c^{(i)}\}_{i=1}^N$ for a class $c$, we estimate its von Mises-Fisher (vMF) distribution parameters $(\boldsymbol{\mu}_c, \kappa_c)$ via Maximum Likelihood Estimation (MLE). The detailed derivation is provided in Appendix F.

- **MLE of Mean Direction ($\hat{\boldsymbol{\mu}}_c$):** The estimate is the normalized direction of the sum of the embedding vectors.

$$\hat{\boldsymbol{\mu}}_c = \frac{\sum_{i=1}^N \boldsymbol{u}_c^{(i)}}{\left\| \sum_{i=1}^N \boldsymbol{u}_c^{(i)} \right\|_2}. \tag{6}$$

- **MLE of Concentration ($\hat{\kappa}_c$):** The estimate is obtained by computing the empirical mean resultant length, $\bar{R}_c$, which serves as a sufficient statistic for the concentration parameter.

$$\bar{R}_c = \left\| \frac{1}{N} \sum_{i=1}^N \boldsymbol{u}_c^{(i)} \right\|_2 \in [0, 1).$$

The estimate $\hat{\kappa}_c$ is then the unique solution to the equation:

$$A_D(\hat{\kappa}_c) = \bar{R}_c, \qquad \text{where} \quad A_D(\kappa) = \frac{I_{D/2}(\kappa)}{I_{D/2-1}(\kappa)}.$$

Since $A_D(\kappa)$ is a strictly increasing function on $[0, \infty)$ with range $[0, 1)$, the inverse $A_D^{-1}$ is well-defined, yielding a unique solution $\hat{\kappa}_c = A_D^{-1}(\bar{R}_c)$. For simplicity, we replace the true parameters with their MLE and continue to denote them as $(\boldsymbol{\mu}_c, \kappa_c)$.

## 3.3 THE VDSA OPTIMIZATION OBJECTIVE

We represent each class $c$ by a vMF distribution $p(\mathbf{u}_c \mid \boldsymbol{\mu}_c, \kappa_c)$ on the unit hypersphere and define the stochastic logit

$$S_c = s\, \mathbf{z}^\top \mathbf{u}_c, \qquad \mathbf{u}_c \sim \text{vMF}(\boldsymbol{\mu}_c, \kappa_c), \ \|\mathbf{z}\|_2 = \|\boldsymbol{\mu}_c\|_2 = 1.$$

The ideal learning objective, therefore, is to optimize the expected cross entropy over the joint distribution of all random prototypes:

$$\mathcal{L}_{\text{ideal}} = \mathbb{E}\left[\underbrace{\log \sum_{c=1}^{C} e^{S_c}}_{(A)} - \underbrace{S_y}_{(B)}\right]. \tag{7}$$

**A Jensen upper bound.** Directly computing the expectation of term **(A)** is intractable, as it involves the expectation of a logarithm over a sum of exponentials, for which no closed form solution exists. We therefore derive a tractable upper bound using Jensen's inequality. Let $Y := \sum_{c=1}^{C} e^{S_c} > 0$. Since $\log(\cdot)$ is a concave function,

$$\mathbb{E}\left[\log \sum_c e^{S_c}\right] = \mathbb{E}[\log Y] \leq \log \mathbb{E}[Y] = \log\left(\sum_c \mathbb{E}\big[e^{S_c}\big]\right). \tag{8}$$

Applying this bound to term **(A)**, while noting that the expectation of term **(B)** can be computed directly, yields an upper bound on the ideal objective:

$$\mathcal{L}_{\text{ideal}} \leq \underbrace{\log \sum_{c=1}^{C} \mathbb{E}\big[e^{S_c}\big]}_{(I)} - \underbrace{\mathbb{E}[S_y]}_{(II)}.$$

**Closed forms under vMF.** We derive closed forms for **(I)** and **(II)** by viewing vMF as an exponential family. With natural parameter $\boldsymbol{\theta}_c = \kappa_c \boldsymbol{\mu}_c$, the partition function is

$$Z(\boldsymbol{\theta}) := \int_{\mathcal{S}^{D-1}} \exp(\boldsymbol{\theta}^\top \mathbf{u})\, d\sigma(\mathbf{u}) = \frac{1}{C_D(\|\boldsymbol{\theta}\|_2)},$$

where $C_D(\cdot)$ is the vMF normalizer.

To evaluate the sum inside term **(I)**, we use the moment generating function identity. For any $\mathbf{h} \in \mathbb{R}^D$, $\mathbb{E}_{\mathbf{u} \sim p(\cdot|\boldsymbol{\theta})}\big[e^{\mathbf{h}^\top \mathbf{u}}\big] = Z(\boldsymbol{\theta} + \mathbf{h})/Z(\boldsymbol{\theta})$. Setting $\mathbf{h} = s\mathbf{z}$ gives the expectation of each exponentiated logit:

$$\mathbb{E}\big[e^{S_c}\big] = \mathbb{E}\big[e^{s\,\mathbf{z}^\top \mathbf{u}_c}\big] = \frac{C_D(\kappa_c)}{C_D(\|\kappa_c \boldsymbol{\mu}_c + s\mathbf{z}\|_2)}. \tag{9}$$

For term **(II)**, we compute the expectation of the logit for the ground-truth class $y$. This corresponds to the mean of the vMF distribution scaled by $s\mathbf{z}^\top$:

$$\mathbb{E}[S_y] = s\,\mathbf{z}^\top \mathbb{E}[\mathbf{u}_y] = s\,A_D(\kappa_y)\,\mathbf{z}^\top \boldsymbol{\mu}_y,$$

where $A_D(\kappa) = I_{D/2}(\kappa)/I_{D/2-1}(\kappa)$ is the ratio of modified Bessel functions of the first kind.

**Objective.** Substituting the closed-form results for terms **(I)** and **(II)** into our upper bound yields the final **V**on Mises–Fisher **D**istributional **S**emantic **A**lignment (VDSA) training objective:

$$\mathcal{L}_{\text{VDSA}} = \log\left(\sum_{c=1}^{C} \frac{C_D(\kappa_c)}{C_D(\|\kappa_c \boldsymbol{\mu}_c + s\mathbf{z}\|_2)}\right) - s\,A_D(\kappa_y)\,\mathbf{z}^\top \boldsymbol{\mu}_y. \tag{10}$$

This loss is a differentiable upper bound to the ideal objective in equation 7 that requires no sampling. In the limits, as $\kappa_c \to \infty$ for all $c$, it recovers the standard cross-entropy loss, while for $\kappa_c = 0$ (or $s = 0$), it yields the constant $\log C$.

## 4 EXPERIMENTS

**Experimental Setup** We evaluate VDSA under two standard problem settings: (i) few-shot learning with varying numbers of shots (Section 4.1), and (ii) base-to-novel generalization (Section 4.2). All experiments are built on the open-source BiomedCLIP (Zhang et al., 2023) backbone. Below we describe datasets, evaluation protocols, baselines, and implementation details.

Table 1: Few-shot classification accuracy averaged over 10 biomedical datasets for different shot numbers ($K = 1\sim16$).

| Method | K=1 | K=2 | K=4 | K=8 | K=16 |
|---|---|---|---|---|---|
| **Zero-shot Methods** | | | | | |
| BiomedCLIP | | | 42.38 | | |
| BiomedCLIP + Ensemble | | | 52.14 | | |
| CLIP-Adapter | $45.52 \pm 2.28$ | $44.70 \pm 1.54$ | $45.30 \pm 1.54$ | $46.54 \pm 1.40$ | $48.45 \pm 1.32$ |
| Tip-Adapter | $50.35 \pm 5.03$ | $53.50 \pm 5.39$ | $58.33 \pm 4.93$ | $62.01 \pm 5.76$ | $67.60 \pm 4.44$ |
| Tip-Adapter-F | $52.54 \pm 7.18$ | $54.16 \pm 6.00$ | $62.30 \pm 6.30$ | $68.11 \pm 3.75$ | $72.61 \pm 2.28$ |
| Standard LP | $50.72 \pm 8.06$ | $55.94 \pm 7.50$ | $62.83 \pm 6.63$ | $67.78 \pm 5.04$ | $71.22 \pm 2.83$ |
| LP++ | $50.27 \pm 7.95$ | $55.66 \pm 6.68$ | $61.85 \pm 6.66$ | $66.14 \pm 4.82$ | $70.52 \pm 3.70$ |
| CoOp | $52.59 \pm 6.67$ | $55.71 \pm 4.10$ | $61.35 \pm 3.36$ | $67.74 \pm 3.18$ | $71.48 \pm 2.85$ |
| CoCoOp | $50.88 \pm 4.41$ | $53.91 \pm 5.30$ | $57.63 \pm 4.70$ | $63.15 \pm 3.58$ | $67.51 \pm 2.20$ |
| KgCoOp | $54.31 \pm 4.68$ | $55.79 \pm 5.13$ | $60.92 \pm 3.97$ | $66.00 \pm 2.42$ | $67.71 \pm 1.90$ |
| ProGrad | $53.67 \pm 5.77$ | $56.42 \pm 4.16$ | $62.10 \pm 3.62$ | $67.06 \pm 3.07$ | $69.21 \pm 2.69$ |
| BiomedCoOp | $56.87 \pm 2.53$ | $59.32 \pm 3.80$ | $64.34 \pm 2.47$ | $68.96 \pm 2.77$ | $73.41 \pm 1.68$ |
| **VDSA (Ours)** | $\mathbf{60.50} \pm 2.06$ | $\mathbf{64.25} \pm 1.75$ | $\mathbf{69.48} \pm 2.50$ | $\mathbf{74.77} \pm 1.20$ | $\mathbf{79.41} \pm 1.28$ |

**Datasets**  VDSA is assessed on ten publicly available biomedical datasets spanning ten organs and eight imaging modalities: Computerized Tomography (CTKidney (Islam et al., 2022)), Endoscopy (Kvasir (Pogorelov et al., 2017)), Fundus Photography (RETINA (Köhler et al., 2013; Porwal et al., 2018)), Histopathology (LC25000 (Borkowski et al., 2019), CHMNIST (Kather et al., 2016)), Magnetic Resonance Imaging (BTMRI (Masoud, 2021)), Optical Coherence Tomography (OCTMNIST (Kermany et al., 2018)), Ultrasound (BUSI (Al-Dhabyani et al., 2020)), and X-Ray (COVID-QU-Ex (Tahir et al., 2021), KneeXray (Chen, 2018)). This diverse benchmark enables a rigorous evaluation of robustness across heterogeneous biomedical tasks and imaging conditions. Full dataset descriptions and train/val/test splits are provided in Appendix B. **Note** that DermaMNIST (Codella et al., 2019; Tschandl et al., 2018) was excluded from the comparison due to class imbalance, which led to BiomedCoOp achieving unnaturally high scores by predicting the dominant class. Further discussion on this issue can be found in the Appendix C.

**Evaluation Protocols**  We adopt two challenging protocols:

**Few-shot Learning.**  To emulate limited data scenarios, models are trained using $K \in \{1, 2, 4, 8, 16\}$ labeled examples per class. This directly tests the sample efficiency of the adaptation method.

**Base-to-Novel Generalization.** Each dataset's classes are partitioned into disjoint base and novel sets. Models are trained on 16-shot samples from base classes and evaluated on both base and unseen novel classes, measuring the ability to retain knowledge while adapting, a key indicator against catastrophic forgetting.

**Baselines**  We compare VDSA with three major families of lightweight adaptation approaches: (1) *Prompt learning methods*: CoOp (Zhou et al., 2022b), CoCoOp (Zhou et al., 2022a), ProGrad (Zhu et al., 2023), KgCoOp (Yao et al., 2023) and BioMedCoOp (Koleilat et al., 2025); (2) *Adapter-based methods*: CLIP-Adapter (Gao et al., 2024), Tip-Adapter (Zhang et al., 2021), and Tip-Adapter-F (Zhang et al., 2021); (3) *Linear probing methods*: standard Linear Probing and LP++ (Huang et al., 2024b). Zero-shot and LLM-prompted zero-shot BiomedCLIP (Zhang et al., 2023) serve as strong reference points. All baselines share the same BiomedCLIP (Zhang et al., 2023) backbone and comparable parameter budgets.

**Implementation Details**  We build on the official BiomedCLIP (Zhang et al., 2023) codebase with a ViT-B/16 vision encoder. For VDSA, class-wise semantic distributions are estimated from $N = 50$ GPT-4 (Achiam et al., 2023) generated prompts and kept fixed during training. Only the LoRA modules in the vision encoder are optimized using SGD with a learning rate of $1 \times 10^{-4}$ and a batch size of 4. Few-shot experiments run for 100 epochs and base-to-novel for 50 epochs. Results are averaged over three random seeds. All experiments are performed on a single NVIDIA RTX 4090 GPU (24 GB VRAM).

Table 2: Performance comparison on 9 biomedical datasets. Bold indicates the best in each row.

| Dataset | | BiomedCLIP | CoOp | CoCoOp | KgCoOp | ProGrad | BiomedCoOp | VDSA (Ours) |
|---|---|---|---|---|---|---|---|---|
| Average on 9 datasets | Base | 49.27 | 76.71 | 75.52 | 71.91 | 75.69 | 78.60 | **82.48** |
| | Novel | 67.17 | 65.34 | 67.74 | 65.94 | 67.33 | 73.90 | **80.12** |
| | HM | 55.23 | 68.80 | 69.11 | 67.22 | 69.86 | 74.04 | **78.55** |
| BTMRI | Base | 40.88 | 82.25 | 77.88 | 78.03 | 82.13 | 82.42 | **86.25** |
| | Novel | 96.18 | 94.51 | 94.84 | 95.05 | 94.98 | **96.84** | 94.66 |
| | HM | 57.37 | 87.95 | 85.53 | 85.70 | 88.09 | 89.05 | **90.26** |
| COVID-QU-Ex | Base | 53.96 | 75.92 | **77.28** | 75.42 | 75.19 | 75.91 | 76.57 |
| | Novel | 89.43 | 90.07 | 87.61 | 89.61 | 90.34 | **91.63** | 91.57 |
| | HM | 67.31 | 82.39 | 82.12 | 81.90 | 82.07 | 83.03 | **83.40** |
| CTKIDNEY | Base | 38.55 | 82.24 | 81.96 | 81.67 | 83.86 | 86.93 | **89.73** |
| | Novel | 52.99 | 67.92 | 56.56 | 58.45 | 63.01 | 78.94 | **85.79** |
| | HM | 44.63 | 74.40 | 66.93 | 68.14 | 71.96 | 82.74 | **87.72** |
| Kvasir | Base | 75.00 | 86.22 | 85.94 | 81.56 | 82.89 | 86.50 | **88.00** |
| | Novel | 60.50 | 58.06 | 53.95 | 59.00 | 60.45 | 61.83 | **69.50** |
| | HM | 66.97 | 69.39 | 66.29 | 68.47 | 69.91 | 72.11 | **77.66** |
| CHMNIST | Base | 37.63 | 89.41 | 87.77 | 75.45 | 82.98 | 88.87 | **92.42** |
| | Novel | 40.69 | 35.11 | 42.51 | 38.70 | 44.19 | 42.73 | **53.06** |
| | HM | 39.10 | 50.42 | 57.28 | 51.16 | 57.67 | 57.71 | **67.42** |
| LC25000 | Base | 59.73 | 90.12 | 88.33 | 88.13 | 90.29 | 93.77 | **95.09** |
| | Novel | 87.60 | 87.55 | 95.02 | 86.44 | 85.47 | **97.00** | 96.27 |
| | HM | 71.03 | 88.82 | 91.55 | 87.28 | 87.81 | 95.36 | **95.68** |
| RETINA | Base | 45.18 | 70.98 | 66.88 | 60.77 | 68.77 | 68.46 | **80.57** |
| | Novel | 55.28 | 56.90 | 65.56 | 54.91 | 58.43 | 67.72 | **74.02** |
| | HM | 49.72 | 63.16 | 66.21 | 57.69 | 63.18 | 68.09 | **79.06** |
| KneeXray | Base | 35.89 | 38.28 | 34.08 | 37.94 | 40.88 | 44.23 | **48.26** |
| | Novel | 71.90 | 47.69 | 63.14 | 61.19 | 59.12 | 78.35 | **81.02** |
| | HM | 47.88 | 42.47 | 44.27 | 46.84 | 48.34 | 56.54 | **60.49** |
| OCTMNIST | Base | 56.60 | 75.00 | 79.60 | 68.20 | 74.20 | 80.33 | **83.40** |
| | Novel | 50.00 | 50.23 | 50.47 | 50.13 | 50.02 | 50.07 | **56.20** |
| | HM | 53.10 | 60.17 | 61.77 | 57.79 | 59.76 | 61.69 | **67.15** |

## 4.1 FEW-SHOT ADAPTATION

Table 1 summarizes few-shot classification accuracy averaged over 10 biomedical datasets. Our proposed **VDSA** consistently achieves the highest accuracy across all shot counts. In the most challenging **1-shot** and **2-shot** settings, VDSA reaches **60.50**% and **64.25**% accuracy, respectively, outperforming the strongest baseline *BiomedCoOp* (56.87%, 59.32%) by **3.6** and **4.9** percentage points, and surpassing other prompt-tuning methods such as CoOp (52.59%, 55.71%) and ProGrad (53.67%, 56.42%) by even larger margins. As the number of labeled examples grows, the advantage of VDSA remains clear: it attains **69.48**%, **74.77**%, and **79.41**% for $K = 4, 8, 16$, consistently exceeding BiomedCoOp (64.34%, 68.96%, 73.41%) and other baselines. The detailed per-dataset performance curves are provided in Appendix D. Across 10 diverse biomedical datasets covering eight imaging modalities, VDSA achieves the best or tied-best performance at nearly every shot count. These strong and uniform gains highlight the robustness of VDSA to both domain variation and data scarcity.

The key to this improvement lies in VDSA's distributional semantic alignment. Instead of compressing multiple prompts into a prototype, VDSA models the entire von Mises–Fisher distribution of class semantics. This richer representation provides an implicit, infinite sample semantic augmentation that optimize the vision–language alignment, maintaining effectiveness as supervision increases.

## 4.2 BASE-TO-NOVEL GENERALIZATION

We next evaluate VDSA on the challenging *Base-to-Novel Generalization* protocol. Table 2 presents base, novel, and harmonic mean (**HM**) accuracies across nine datasets and their overall averages, with HM reflecting the level of balanced generalization. Note that the BUSI Al-Dhabyani et al. (2020) dataset is excluded here since it contains only three classes, making a base–novel split infeasible. VDSA consistently ranks among the best on Base, Novel, and HM metrics, with gains exceeding 10% in HM on representative datasets such as CHMNIST Kather et al. (2016) and RETINA Köhler et al. (2013); Porwal et al. (2018). These results demonstrate that modeling each class as a von

Table 3: Ablation study of VDSA. The few-shot part reports average accuracy (%) over 10 biomedical datasets under different shots (1, 2, 4, 8, 16). The base-to-novel part reports average Base, Novel, and their harmonic mean (HM) accuracies (%).

| Method | Few-shot Avg. Accuracy (%) | | | | | Base-to-Novel Avg. (%) | | |
|---|---|---|---|---|---|---|---|---|
| | 1-shot | 2-shot | 4-shot | 8-shot | 16-shot | Base | Novel | HM |
| CE (w/o VDSA) | 55.77 | 58.72 | 66.50 | 72.37 | 78.19 | 81.26 | 77.50 | 77.51 |
| **VDSA (Ours)** | **60.50** | **64.25** | **69.48** | **74.77** | **79.41** | **82.48** | **80.12** | **78.55** |

Mises–Fisher semantic distribution, rather than a single prototype, mitigates overfitting to base concepts and enhances transfer to unseen classes.

### 4.3 PARAMETERS SENSITIVITY

We analyze the impact of the number of prompts in Fig. 2. In our formulation, prompts are regarded as samples from the latent semantic distribution of each class, and the number of prompts affects the reliability of the vMF parameter estimation. Increasing prompts from 1 to 20 leads to a marked improvement in novel-class and HM accuracy. Beyond 20 prompts, performance changes marginally: base accuracy stays around 82.5%–82.7%, and novel accuracy grows slightly to 80.12% at 50 prompts. These results indicate that using about 30 prompts already captures most of the achievable gains, and adding more prompts provides marginal additional benefit.

### 4.4 ABLATION STUDY

To evaluate the contribution of the proposed VDSA module, we conduct an ablation study by removing VDSA and training the model with the conventional cross entropy classification loss. This reduces the framework to a prototype-based baseline, where images are aligned only with the mean text embedding of each class rather than with a full semantic distribution. Table 3 reports the average results for base-to-novel generalization, highlighting the consistent improvements brought by VDSA. Across all shot levels from 1 to 16, VDSA consistently surpasses the CE baseline. The improvements are most evident in the low-shot regime, with gains of +4.7% at 1-shot and +5.5% at 2-shot, showing that representing each class as a von Mises–Fisher semantic dis-

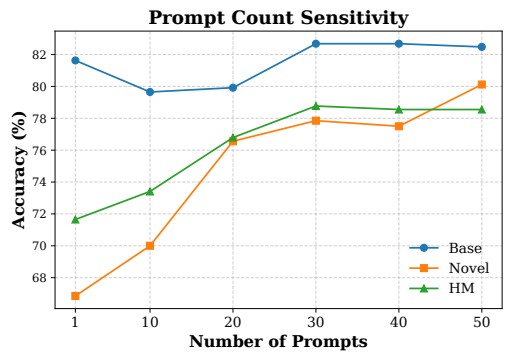

Figure 2: Effect of the Number of Prompts on base-to-novel generalization.

tribution leads to stronger and more stable few-shot performance. Under the base-to-novel protocol, VDSA also achieves consistent gains. The average HM rises from 77.51% to 78.55%, reflecting better balance between base and novel classes and stronger transfer to unseen categories.

## 5 CONCLUSION

In this work, we identified a fundamental limitation in existing prompt ensembling methods for Vision-Language Models: the collapse of rich semantic diversity into a single prototype. To address this, we introduced VDSA, a novel parameter-efficient fine-tuning framework that shifts the paradigm from prototype alignment to distributional alignment. By modeling class semantics with the von Mises-Fisher distribution and deriving a tractable upper-bound objective, VDSA learns to align image features with the entire semantic space of the concepts. Our extensive experiments on 10 biomedical datasets demonstrate that this principled approach yields significant improvements in both few-shot adaptation and base-to-novel generalization, establishing a new state-of-the-art for adapting VLMs to specialized domains.

ETHICS STATEMENT

This study utilizes publicly available biomedical datasets, all of which have been anonymized to ensure privacy and comply with ethical standards. We adhered to ethical guidelines for data usage and have taken measures to avoid potential harmful insights. There are no conflicts of interest, and all research practices were in accordance with legal and ethical research standards.

REPRODUCIBILITY STATEMENT

To ensure the reproducibility of our results, we will provide the source code and datasets upon publication. All theoretical results, assumptions, and proofs are detailed in the paper. Data processing steps necessary for reproducing the experiments are also included.

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

APPENDIX

## A   LLM USAGE

In this work, we use GPT-5 to assist in refining and polishing the paper. In addition, we use GPT-4 (Achiam et al., 2023) to generate textual descriptions for each medical class in the datasets. Specifically, for each dataset with $C$ classes, we queried GPT-4 with the following instruction:

> Give $N$ textual descriptions of visual discriminative features for distinct medical cases of [CLASS] found in [MODALITY].

These generated descriptions were then used as prompts in the subsequent model training and evaluation process. Here we include one representative text prompt for each class across all datasets.

**BTMRI**

- normal brain: A normal brain in MRI appears with clearly defined structures, no abnormal growths, and symmetrical hemispheres.
- glioma tumor: A glioma tumor in MRI appears as an irregular, heterogeneous mass with poorly defined borders.
- meningioma tumor: A meningioma tumor in MRI appears as a well-circumscribed, extra-axial mass with a broad dural attachment.
- pituitary tumor: A pituitary tumor in MRI appears as a well-defined sellar mass, often with suprasellar extension.

**BUSI**

- benign tumor: an ultrasound image showing well-defined, smooth margins, indicating a benign breast tumor.
- malignant tumor: an ultrasound image showing irregular or spiculated margins, indicating a malignant breast tumor.
- normal scan: an ultrasound image showing homogenous echotexture throughout, indicating a normal breast ultrasound scan.

**CTKidney**

- cyst kidney: a CT image showing a well-circumscribed lesion with thin, smooth walls, suggestive of a cyst kidney.
- kidney stone: a CT image showing a hyperdense focus within the renal pelvis, indicating a kidney stone.
- kidney tumor: a CT image showing a solid mass with heterogeneous enhancement, indicating a kidney tumor.
- normal kidney: a CT image showing well-defined renal contours and normal cortical thickness, indicating a normal kidney.

**COVID-QU-Ex**

- covid lungs: an X-ray image showing bilateral ground-glass opacities, indicating covid lungs.
- lung opacity lungs: an X-ray image showing localized ground-glass opacity, indicating lung opacity lungs.
- normal lungs: an X-ray image showing clear lung fields, indicating normal lungs.
- viral pneumonia lungs: an X-ray image showing bilateral ground-glass opacities, indicating viral pneumonia lungs.

**Kvasir**

- dyed lifted polyps: Dyed lifted polyps appear as raised lesions with a blue or green hue, indicating the application of dye to enhance visibility during endoscopy.

- dyed resection margins: Dyed resection margins appear as clearly delineated edges around a resected lesion, marked by the application of dye during endoscopy.

- esophagitis: Esophagitis presents as inflamed, reddened areas of the esophageal lining, often with visible erosions or ulcers in severe cases.

- normal cecum: The normal cecum appears as a blind-ended pouch located at the beginning of the large intestine, with a smooth and healthy mucosal surface.

- normal pylorus: The normal pylorus appears as a circular, well-defined opening at the distal end of the stomach, leading into the duodenum.

- normal z line: The normal Z line appears as a distinct, zigzagging line at the junction of the esophagus and stomach during endoscopy.

- polyps: Polyps appear as protrusions or growths on the mucosal surface of the gastrointestinal tract during endoscopy.

- ulcerative colitis: Ulcerative colitis appears as inflammatory changes in the mucosal lining of the colon during endoscopy.

**KneeXray**

- healthy knee: An X-ray image showing a knee with a clear and even joint space, no bone spurs, indicating a healthy knee.

- doubtful osteoarthritis: An X-ray image showing a knee with slight joint space narrowing and the beginning of osteophyte formation, indicating doubtful osteoarthritis.

- minimal osteoarthritis: An X-ray image showing a knee with definite osteophytes and slight joint space narrowing, indicating minimal osteoarthritis.

- moderate osteoarthritis: An X-ray image showing a knee with multiple, well-defined osteophytes and moderate joint space narrowing, indicating moderate osteoarthritis.

- severe osteoarthritis: An X-ray image showing a knee with large osteophytes, severe joint space narrowing, and significant sclerosis, indicating severe osteoarthritis.

**RETINA**

- cataract: a retina image showing opacification of the lens with loss of transparency, indicative of cataract.

- diabetic retinopathy: a retina image showing microaneurysms, hemorrhages, and exudates in the retina, indicative of diabetic retinopathy.

- glaucoma: a retina image showing optic disc cupping, neuroretinal rim thinning, and retinal nerve fiber layer defects, indicative of glaucoma.

- normal retina: a retina image showing intact retinal layers with well-defined foveal depression and normal vasculature, indicating normal retina.

**CHMNIST**

- adipose tissue: Adipose tissue in histology slides appears as clusters of large, clear cells with a thin rim of cytoplasm and a centrally located nucleus.

- complex stroma: Complex stroma in histology slides presents as a dense network of collagen fibers, fibroblasts, and extracellular matrix components, providing structural support to tissues.

- debris: Debris in histology slides often appears as irregular, darkly staining particles scattered throughout the tissue section, indicating areas of cellular breakdown.

- empty background: Empty background in histology slides appears as clear or lightly stained areas devoid of cellular or extracellular matrix components, providing contrast for identifying tissue structures.
- immune cells: Immune cells in histology slides appear as small, round cells with darkly staining nuclei, often located within or surrounding areas of tissue inflammation.
- normal mucosal glands: Normal mucosal glands in histology slides appear as well-organized tubular structures lined by epithelial cells, often with a central lumen.
- simple stroma: Simple stroma in histology slides consists of a loose network of connective tissue, with sparse collagen fibers and scattered fibroblasts.
- tumour epithelium: Tumor epithelium in histology slides appears as atypical epithelial cells with irregular nuclei, prominent nucleoli, and increased mitotic activity.

**LC25000**

- colon adenocarcinoma: a histopathological section showing malignant glands infiltrating the colonic mucosa and submucosa, indicative of colon adenocarcinoma.
- colon benign tissue: a histological slide showing normal colonic mucosa with intact crypt architecture and absence of dysplastic changes, indicating colon benign tissue.
- lung adenocarcinoma: a histopathological section showing glandular structures infiltrating the lung parenchyma with stromal desmoplasia, indicative of lung adenocarcinoma.
- lung benign tissue: a histological slide showing normal lung parenchyma with intact alveolar architecture and absence of dysplastic changes, indicating lung benign tissue.
- lung squamous cell carcinoma: a histopathological section showing nests of squamous cells with keratinization and intercellular bridges infiltrating the lung tissue, indicative of lung squamous cell carcinoma.

**DermaMNIST**

- actinic keratosis: Actinic keratosis presents as rough, scaly patches on sun-exposed areas of the skin, often with a pink or red base.
- basal cell carcinoma: Basal cell carcinoma (BCC) often appears as a pearly or translucent nodule with visible blood vessels (telangiectasia) on the surface.
- benign keratosis: Benign keratosis, such as seborrheic keratosis, presents as well-defined, warty, or waxy growths on the skin.
- dermatofibroma: Dermatofibromas present as firm, raised nodules on the skin, typically with a hyperpigmented surface.
- melanoma: a skin image showing an irregular, asymmetrical mole, indicating melanoma.
- melanocytic nevus: Melanocytic nevi, or moles, are common benign skin lesions resulting from the proliferation of melanocytes.
- squamous cell carcinoma: a close-up of a lesion with irregular borders and a scaly surface, indicating squamous cell carcinoma.
- vascular lesion: Vascular lesions include a variety of conditions such as hemangiomas, vascular malformations, and pyogenic granulomas.

**OCTMNIST**

- choroidal neovascularization: OCT scan showing subretinal fluid accumulation due to abnormal blood vessels growth.
- diabetic macular edema: OCT image showing retinal thickening with visible cystoid spaces.
- drusen: OCT showing small, hyperreflective deposits beneath the retinal pigment epithelium.
- normal OCT scan: OCT image showing a normal, uniform retinal structure with no signs of fluid accumulation.

## B  DATASET

**Datasets.**  We evaluate on eleven public datasets spanning nine biomedical imaging modalities and ten organs. For each dataset, we state the modality and organ(s), list the class labels as given, and report the train/val/test split.

- **CTKidney** (Islam et al., 2022). *Modality: Computerized Tomography; Organ(s): Kidney*. Classes: Kidney Cyst, Kidney Stone, Kidney Tumor, Normal Kidney. Split: 6221/2487/3738.
- **DermaMNIST** (Codella et al., 2019; Tschandl et al., 2018). *Modality: Dermatoscopy; Organ(s): Skin*. Classes: Actinic Keratosis, Basal Cell Carcinoma, Benign Keratosis, Dermatofibroma, Melanocytic nevus, Melanoma, Vascular Lesion. Split: 7007/1003/2005.
- **Kvasir** (Pogorelov et al., 2017). *Modality: Endoscopy; Organ(s): Colon*. Classes: Dyed Lifted Polyps, Normal Cecum, Esophagitis, Dyed Resection Margins, Normal Pylorus, Normal Z Line, Polyps, Ulcerative Colitis. Split: 2000/800/1200.
- **RETINA** (Köhler et al., 2013; Porwal et al., 2018). *Modality: Fundus Photography; Organ(s): Retina*. Classes: Cataract, Diabetic Retinopathy, Glaucoma, Normal Retina. Split: 2108/841/1268.
- **LC25000** (Borkowski et al., 2019). *Modality: Histopathology; Organ(s): Lung, Colon*. Classes: Colon Adenocarcinoma, Colon Benign Tissue, Lung Adenocarcinoma, Lung Benign Tissue, Lung Squamous Cell Carcinoma. Split: 12500/5000/7500.
- **CHMNIST** (Kather et al., 2016). *Modality: Histopathology; Organ(s): Colorectal*. Classes: Adipose Tissue, Complex Stroma, Debris, Empty Background, Immune Cells, Normal Mucosal Glands, Simple Stroma, Tumor Epithelium. Split: 2496/1000/1504.
- **BTMRI** (Masoud, 2021). *Modality: Magnetic Resonance Imaging; Organ(s): Brain*. Classes: Glioma Tumor, Meningioma Tumor, Normal Brain, Pituitary Tumor. Split: 2854/1141/1717.
- **OCTMNIST** (Kermany et al., 2018). *Modality: Optical Coherence Tomography; Organ(s): Retina*. Classes: Choroidal Neovascularization, Drusen, Diabetic Macular Edema, Normal. Split: 97477/10832/1000.
- **BUSI** (Al-Dhabyani et al., 2020). *Modality: Ultrasound; Organ(s): Breast*. Classes: Benign Tumors, Malignant Tumors, Normal Scans. Split: 389/155/236.
- **COVID-QU-Ex** (Tahir et al., 2021). *Modality: X-Ray; Organ(s): Chest*. Classes: COVID-19, Lung Opacity, Normal Lungs, Viral Pneumonia. Split: 10582/4232/6351.
- **KneeXray** (Chen, 2018). *Modality: X-Ray; Organ(s): Knee*. Classes: No, Doubtful, Minimal, Moderate, and Severe Osteoarthritis. Split: 5778/826/1656.

## C  DERMAMNIST PERFORMANCE

Table 4 reports results on the 7-class DermaMNIST test set (2,005 images), which is highly imbalanced: the majority class *melanocytic nevus* constitutes 1,341 samples (66.9%). Under this skew, plain accuracy is unreliable because a trivial majority class predictor already attains 66.9%.

**Metrics.**  We report (i) *Macro-Recall* (balanced accuracy), (ii) *Macro-F1*, (iii) the *Zero-Recall (count)*—the number of classes with recall exactly 0, and (iv) *Minority Avg. Recall*, the mean recall over the other 6 classes. All values are shown as percentages except the count.

**Findings.**  At 1-shot, although BiomedCoOp reaches 61.7% accuracy, it still trails the 66.9% majority baseline and exhibits poor balance (Macro-Recall 25.2%, Macro-F1 19.1%, two zero-recall classes). Our method (1-shot) improves Macro-Recall/F1 to 30.5%/21.6%, removes zero-recall classes, and nearly doubles minority class recall (14.6% → 28.9%). At 16-shot, BiomedCoOp modestly improves (Macro-Recall 29.7%, Macro-F1 26.5%) but still has one zero-recall class and only 21.0% minority recall. In contrast, Ours (16-shot) delivers substantially better balance (Macro-Recall 54.6%, Macro-F1 39.1%, zero-recall count 0, minority recall 52.3%). While its accuracy

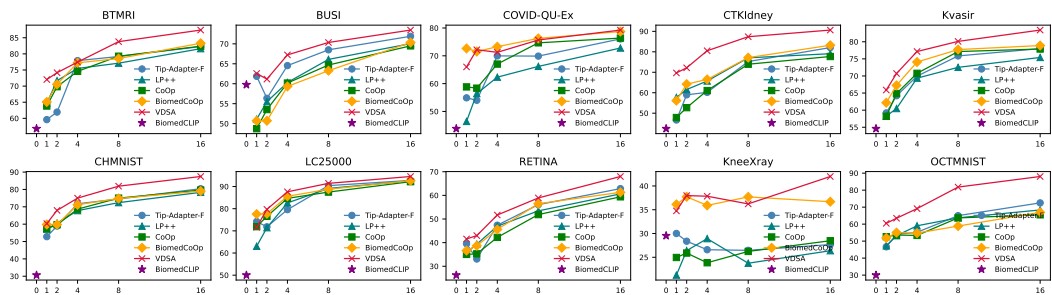

Figure 3: Per-dataset few-shot classification performance across 10 biomedical datasets.

(59.8%) is slightly below BiomedCoOp (61.5%), both are inferior to the 66.9% majority baseline underscoring that balanced metrics, not accuracy, reflect clinically meaningful performance under severe imbalance.

Table 4: DermaMNIST under severe class imbalance (values in %).

| Metric | Majority – | BiomedCoOp 1-shot | BiomedCoOp 16-shot | Ours 1-shot | Ours 16-shot |
|---|---|---|---|---|---|
| Accuracy (%) | **66.9** | 61.7 | 61.5 | 36.3 | 59.8 |
| Macro-Recall (%) | – | 25.2 | 29.7 | 30.5 | **54.6** |
| Macro-F1 (%) | – | 19.1 | 26.5 | 21.6 | **39.1** |
| Zero-Recall (count) | – | 2 | 1 | **0** | **0** |
| Minority Avg. Recall (%) | – | 14.6 | 21.0 | 28.9 | **52.3** |

## D PER-DATASET FEW-SHOT PERFORMANCE

Figure 3 reports the per-dataset few-shot classification performance across 10 biomedical datasets under different shot numbers ($K = 1 \sim 16$). Consistent with the averaged results in the main text, VDSA attains the best or tied-best accuracy across nearly all datasets and shot counts, further confirming its robustness to diverse imaging modalities and limited-data settings.

## E LIMITATION OF GAUSSIAN-BASED DISTRIBUTION

**Setup.** Let image features be $\ell_2$-normalized, $\|\mathbf{z}\|_2 = 1$, and each class $c$ be represented by a random prototype $\mathbf{u}_c$. We compare (i) an *Euclidean Gaussian* model $\mathbf{u}_c \sim \mathcal{N}(\boldsymbol{\mu}_c, \boldsymbol{\Sigma}_c)$ in $\mathbb{R}^D$ with (ii) an *on-sphere* model. We study the Jensen upper bound

$$\mathcal{L} = \log\sum_{c=1}^{C} \mathbb{E}\Big[e^{\,s\,\mathbf{z}^\top \mathbf{u}_c}\Big] \;-\; \mathbb{E}\big[s\,\mathbf{z}^\top \mathbf{u}_y\big], \qquad s > 0.$$

A. EUCLIDEAN GAUSSIAN VIOLATES THE UNIT-SPHERE COSINE GEOMETRY

**Closed form under Gaussian.** If $\mathbf{u}_c \sim \mathcal{N}(\boldsymbol{\mu}_c, \boldsymbol{\Sigma}_c)$ in $\mathbb{R}^D$, then $\mathbf{z}^\top \mathbf{u}_c \sim \mathcal{N}(\mathbf{z}^\top \boldsymbol{\mu}_c, \mathbf{z}^\top \boldsymbol{\Sigma}_c \mathbf{z})$. Hence

$$\mathbb{E}\Big[e^{\,s\,\mathbf{z}^\top \mathbf{u}_c}\Big] = e^{\,s\,\mathbf{z}^\top \boldsymbol{\mu}_c + \frac{s^2}{2}\mathbf{z}^\top \boldsymbol{\Sigma}_c \mathbf{z}}, \quad \mathbb{E}\big[s\,\mathbf{z}^\top \mathbf{u}_y\big] = s\,\mathbf{z}^\top \boldsymbol{\mu}_y,$$

and therefore

$$\mathcal{L}_{\text{Gauss}} = \log\sum_{c=1}^{C} e^{\,s\,\mathbf{z}^\top \boldsymbol{\mu}_c + \frac{s^2}{2}\mathbf{z}^\top \boldsymbol{\Sigma}_c \mathbf{z}} \;-\; s\,\mathbf{z}^\top \boldsymbol{\mu}_y. \tag{11}$$

**Geometric mismatch.** Differentiating equation 11 w.r.t. $\mathbf{z}$ gives

$$\nabla_{\mathbf{z}}\mathcal{L}_{\text{Gauss}} = \sum_{c=1}^{C} w_c\left(s\,\boldsymbol{\mu}_c + s^2\,\boldsymbol{\Sigma}_c\mathbf{z}\right) \; - \; s\,\boldsymbol{\mu}_y, \quad w_c = \frac{e^{a_c}}{\sum_j e^{a_j}}, \;\; a_c = s\,\mathbf{z}^{\top}\boldsymbol{\mu}_c + \frac{s^2}{2}\,\mathbf{z}^{\top}\boldsymbol{\Sigma}_c\mathbf{z}.$$

Compared with the cosine geometry objective (which depends only on angles $\mathbf{z}^{\top}\boldsymbol{\mu}_c$), the Gaussian bound introduces the variance driven term $s^2\,\boldsymbol{\Sigma}_c\mathbf{z}$. It biases $\mathbf{z}$ toward directions of small $\mathbf{z}^{\top}\boldsymbol{\Sigma}_c\mathbf{z}$ (low variance axes), making optimization depend on scale/variance in $\mathbb{R}^D$ rather than purely on directions on the unit sphere. This contradicts CLIP's spherical cosine geometry.

## B. PROJECTED GAUSSIAN ON THE SPHERE IS NOT AN EXPONENTIAL FAMILY

**Projected (normalized) Gaussian.** An on-sphere alternative is to sample $\mathbf{g} \sim \mathcal{N}(\boldsymbol{\mu}, \boldsymbol{\Sigma})$ and project $\mathbf{u} = \mathbf{g}/\|\mathbf{g}\| \in \mathcal{S}^{D-1}$. The induced density on the sphere (with respect to surface measure $d\sigma$) is

$$p_{\text{PN}}(\mathbf{u} \mid \boldsymbol{\mu}, \boldsymbol{\Sigma}) \;\propto\; \int_0^{\infty} r^{D-1}\, e^{-\frac{1}{2}\,(r\mathbf{u}-\boldsymbol{\mu})^{\top}\boldsymbol{\Sigma}^{-1}(r\mathbf{u}-\boldsymbol{\mu})}\, dr. \tag{12}$$

Expanding the quadratic form inside the integral yields dependence on both $\mathbf{u}^{\top}\boldsymbol{\Sigma}^{-1}\mathbf{u}$ and $\mathbf{u}^{\top}\boldsymbol{\Sigma}^{-1}\boldsymbol{\mu}$ in a *nonlinear* fashion after integrating out $r$.

**Consequence for the Jensen bound.** The vMF density is an exponential family on $\mathcal{S}^{D-1}$: $p_{\text{vMF}}(\mathbf{u} \mid \boldsymbol{\theta}) \propto e^{\boldsymbol{\theta}^{\top}\mathbf{u}}$, which implies the key identity

$$\mathbb{E}_{\text{vMF}(\boldsymbol{\theta})}\left[e^{\mathbf{h}^{\top}\mathbf{u}}\right] = \frac{Z(\boldsymbol{\theta}+\mathbf{h})}{Z(\boldsymbol{\theta})}.$$

In contrast, equation 12 cannot be written in the form $p(\mathbf{u} \mid \boldsymbol{\theta}) = e^{\boldsymbol{\theta}^{\top}\mathbf{u}-A(\boldsymbol{\theta})}$ with respect to $d\sigma$ for general $(\boldsymbol{\mu}, \boldsymbol{\Sigma})$. Therefore the above identity is unavailable, and the expectation $\mathbb{E}\left[e^{s\,\mathbf{z}^{\top}\mathbf{u}}\right]$ under the projected Gaussian does not reduce to a tractable ratio of normalizers as in vMF. Hence one cannot obtain an analytic Jensen-type upper bound of the same closed form used in our method.

**Summary.** Euclidean Gaussians introduce variance sensitive terms that break spherical cosine geometry; projected Gaussians live on the sphere but lack the exponential-family structure needed to turn $\mathbb{E}\left[e^{s\,\mathbf{z}^{\top}\mathbf{u}}\right]$ into a closed-form expression. The vMF model satisfies both requirements simultaneously.

## F   DERIVATION OF vMF MAXIMUM LIKELIHOOD ESTIMATORS

Here, we derive the maximum likelihood estimators (MLE) for the parameters of the von Mises-Fisher (vMF) distribution, given $N$ i.i.d. samples $\{\boldsymbol{u}_i\}_{i=1}^{N}$ from vMF$(\boldsymbol{\mu}, \kappa)$. The probability density function is:

$$p(\boldsymbol{u} \mid \boldsymbol{\mu}, \kappa) = C_D(\kappa)\exp(\kappa\boldsymbol{\mu}^{\top}\boldsymbol{u}),$$

where $C_D(\kappa)$ is the normalization constant. The log-likelihood function $\mathcal{L}$ for the $N$ samples is:

$$\log\mathcal{L}(\boldsymbol{\mu}, \kappa) = \log\prod_{i=1}^{N} p(\boldsymbol{u}_i \mid \boldsymbol{\mu}, \kappa) \tag{13}$$

$$= \sum_{i=1}^{N} \log\left[C_D(\kappa)\exp(\kappa\boldsymbol{\mu}^{\top}\boldsymbol{u}_i)\right] \tag{14}$$

$$= N\log C_D(\kappa) + \kappa\sum_{i=1}^{N}\boldsymbol{\mu}^{\top}\boldsymbol{u}_i \tag{15}$$

$$= N\log C_D(\kappa) + \kappa\boldsymbol{\mu}^{\top}\left(\sum_{i=1}^{N}\boldsymbol{u}_i\right). \tag{16}$$

**Estimating Mean Direction $\hat{\mu}$.** To maximize the log-likelihood, we first focus on the term involving $\boldsymbol{\mu}$, which is $\kappa\boldsymbol{\mu}^\top\left(\sum_{i=1}^N \boldsymbol{u}_i\right)$. Let the resultant vector be $\mathbf{R} = \sum_{i=1}^N \boldsymbol{u}_i$. Since $\kappa \geq 0$, maximizing this term is equivalent to maximizing the dot product $\boldsymbol{\mu}^\top\mathbf{R}$. By the Cauchy-Schwarz inequality, this dot product is maximized when the unit vector $\boldsymbol{\mu}$ is aligned with the vector $\mathbf{R}$. Therefore, the MLE for the mean direction is the normalized resultant vector:

$$\hat{\boldsymbol{\mu}} = \frac{\mathbf{R}}{\|\mathbf{R}\|_2} = \frac{\sum_{i=1}^N \boldsymbol{u}_i}{\left\|\sum_{i=1}^N \boldsymbol{u}_i\right\|_2}.$$

**Estimating Concentration $\hat{\kappa}$.** We substitute the MLE $\hat{\boldsymbol{\mu}}$ back into the log-likelihood function (Eq. 16). The second term becomes:

$$\kappa\hat{\boldsymbol{\mu}}^\top\mathbf{R} = \kappa\frac{\mathbf{R}^\top}{\|\mathbf{R}\|_2}\mathbf{R} = \kappa\frac{\|\mathbf{R}\|_2^2}{\|\mathbf{R}\|_2} = \kappa\|\mathbf{R}\|_2.$$

The log-likelihood is now a function of $\kappa$ alone:

$$\log\mathcal{L}(\kappa) = N\log C_D(\kappa) + \kappa\|\mathbf{R}\|_2.$$

To find the maximum, we differentiate with respect to $\kappa$ and set the result to zero:

$$\frac{\partial\log\mathcal{L}}{\partial\kappa} = N\frac{C_D'(\kappa)}{C_D(\kappa)} + \|\mathbf{R}\|_2 = 0.$$

Using the known identity for vMF distributions, $A_D(\kappa) = -\frac{d}{d\kappa}\log C_D(\kappa) = -\frac{C_D'(\kappa)}{C_D(\kappa)}$, we have:

$$N(-A_D(\kappa)) + \|\mathbf{R}\|_2 = 0.$$

Rearranging gives the final equation for the MLE $\hat{\kappa}$:

$$A_D(\hat{\kappa}) = \frac{\|\mathbf{R}\|_2}{N} = \frac{\left\|\sum_{i=1}^N \boldsymbol{u}_i\right\|_2}{N} = \left\|\frac{1}{N}\sum_{i=1}^N \boldsymbol{u}_i\right\|_2.$$

This shows that the MLE for the concentration $\hat{\kappa}$ is the solution to $A_D(\hat{\kappa}) = \bar{R}$, where $\bar{R}$ is the empirical mean resultant length.

