# OpenReview forum: "vMF Guided Learning for Biomedical Vision-Language Models"
_ICLR.cc/2026/Conference — ICLR 2026 Conference Withdrawn Submission_

### Official Review · Reviewer_1rU4 · 2025-10-24

**Soundness:** 3
**Presentation:** 3
**Contribution:** 3
**Rating:** 6
**Confidence:** 3

**Summary:**

This paper proposes vMF Distribution Semantic Alignment (VDSA), a new method for adapting biomedical Vision–Language Models (VLMs) by modeling prompt ensembles as distributions rather than point prototypes. The key idea is to represent each class’s textual embeddings as samples from a von Mises–Fisher (vMF) distribution on the unit hypersphere, capturing both the semantic center and its dispersion. The authors derive a closed-form, differentiable upper bound of the expected contrastive loss, avoiding sampling while effectively aligning image embeddings to the full semantic distribution. The method is applied to BiomedCLIP, fine-tuned with LoRA on the vision encoder, and evaluated on a wide range of biomedical benchmarks under both few-shot and base-to-novel generalization settings. Results show consistent improvements across 10 datasets, achieving new state-of-the-art performance in most cases.

**Strengths:**

- **Mathematically grounded formulation.** The vMF-based modeling and Jensen upper bound derivation are clearly motivated and theoretically sound.
- **Comprehensive evaluation.** The experiments span 10 biomedical datasets and both few-shot and base-to-novel protocols. The results are consistent and convincing.
- **Clarity and presentation.** The paper is well-written and easy to follow.

**Weaknesses:**

- **Marginal empirical gain in the ablation study.** While Table 3 confirms the benefit of the VDSA loss over standard CE, the improvement is somewhat modest. When comparing to other baselines in Table 1 and Table 2, the ablated variant with standard CE still outperforms most of them. That is, currently it seems the major factor of improved performance comes from a better design choice of PEFT method (LoRA on vision encoder).
 To strengthen the empirical claim, it would be helpful to repeat the ablation experiment under different PEFT design choices (e.g., different LoRA config, adapter tuning, or bias tuning) to confirm that the benefit of VDSA is consistent regardless of the adaptation mechanism.
- **Lack of qualitative prompt analysis.** Since the central motivation is to preserve the semantic diversity of the LLM-generated prompt set, the paper would benefit from a qualitative diversity analysis. For example, the authors could visualize prompts that are close to or far from the mean embedding.

    Additionally, a small case study comparing examples from Table 3 (with/without VDSA) could provide interpretability evidence that VDSA helps recover instances benefiting from distributional modeling.

- **Limited discussion of generality.** The paper focuses solely on biomedical data, but the proposed formulation seems domain-agnostic. It would be useful to clarify whether VDSA is also expected to improve performance on general-domain data, or if the gains are specific to domain gaps like biomedical semantics.

**Questions:**

- Why does the paper restrict the scope to the biomedical domain? Is there any empirical observation or explanation for why VDSA might not benefit general-domain VLMs?
- It would be helpful if the authors could provide more insights into the LLM component used to generate diverse textual descriptions:
    - Did the authors experiment with different prompting strategies or instruction styles (e.g., using few-shot examples) when querying GPT-4?
    - How well do these generated descriptions align with the BiomedCLIP training distribution?

---

### Official Review · Reviewer_XuKw · 2025-10-27

**Soundness:** 2
**Presentation:** 2
**Contribution:** 2
**Rating:** 2
**Confidence:** 3

**Summary:**

This paper proposes a prompt learning method for vision-language models, specifically BiomedCLIP, which learns the distributional information of text embedding. The method assumes that normalized text embedding follows a von Mises–Fisher distribution (a Gaussian distribution on the unit hyper-sphere) and then minimizes an upper bound loss for optimization. Experimental results on several biomedical datasets demonstrate the effectiveness of the proposed method.

**Strengths:**

1. Learning the text embeddings with few samples is demonstrated effective in vision-language models on fine-grained datasets. Transfer this idea into biomedical area is straightforward and convincing.

2. Experimental results demonstrate the effectiveness of propose method on several biomedical datasets.

**Weaknesses:**

1. **Limited Novelty.** The primary concern is the novelty of the proposed method. Similar techniques that model the distributional information of text embedding for CLIP models have already been explored, such as in ProDA [1], DAPT [2] and Frolic [3]. Moreover, ProDA proposes a similar theoretical and method framework, which minimizes an upper bound loss to learn distributions of text embedding. The paper should clarify what is the differences between proposed method and prior works.

2. **Lack of ablation study.** This paper assuming the normalized text embedding follows a von Mises–Fisher distribution and then minimize an upper bound loss for optimization. However, a simple baseline method is the original text embeddings follow a Gaussian distribution, as discussed in ProDA. An ablation study or additional experiments on this baseline would strengthen the justification for the chosen distribution.

3. **Unclear research question.** In the abstract, this paper claims that “current prompt ensembling methods share a fundamental limitation in how they aggregate semantic diversity” (Line 59-60). However, I believe this is not a unique problem for biomedical image classification. I suggest that the author can either refine the research question by focusing on biomedical area or include more diverse benchmarks to demonstrate broader applicability.

[1] Prompt Distribution Learning, CVPR 22

[2] Distribution-Aware Prompt Tuning for Vision-Language Models, ICCV 23

[3] Enhancing Zero-Shot Vision Models by Label-Free Prompt Distribution Learning and Bias Correcting, NeurIPS 24

**Questions:**

See my questions in weakness part.

---

### Official Review · Reviewer_Ei5b · 2025-10-28

**Soundness:** 3
**Presentation:** 3
**Contribution:** 3
**Rating:** 6
**Confidence:** 5

**Summary:**

This paper introduces Von Mises–Fisher Distribution Semantic Alignment (VDSA), a method designed to improve the adaptation of biomedical vision–language models (VLMs). Instead of aggregating multiple prompt embeddings into a single averaged prototype, the authors model class semantics as von Mises–Fisher (vMF) distributions on the hypersphere. They further derive a closed-form Jensen upper bound for the expected contrastive loss, allowing for sampling-free optimization. Experiments show improvements over several strong baselines, particularly in few-shot and base-to-novel generalization settings.

**Strengths:**

1. The idea of using vMF distributions to represent semantic variability is both geometrically appropriate and intuitively motivated, especially since CLIP features are unit-normalized. The derivation of a differentiable, sampling-free upper bound is mathematically neat and avoids common approximations used in probabilistic alignment methods.

2. The paper evaluates across a broad range of biomedical datasets covering different imaging modalities. Improvements over baselines such as BiomedCoOp and CoOp are consistent and not confined to a single dataset or setting.

3. The writing is clear, the mathematical exposition is rigorous but accessible, and the figures effectively convey the main intuition.
Overall, the paper is pleasant to read and easy to follow.

**Weaknesses:**

1. While technically sound, the contribution builds on existing prompt-ensembling and parameter-efficient fine-tuning methods rather than redefining the paradigm. Modeling prototypes as a distribution is a natural extension, but not an entirely new direction. The work feels more like a principled refinement than a novel idea.

2.  The Jensen upper bound is elegant, but the paper does not provide an analysis of how tight the bound is or how it behaves under varying κ values. Without this, it is difficult to assess whether the observed performance gains come from better modeling or from favorable regularization effects.

3. The use of a single vMF per class implicitly assumes that each class’s semantic space is unimodal. In medical imaging, where different anatomical patterns or disease subtypes may exist within a single label, this assumption may not hold. A mixture-based or hierarchical variant could be a more realistic representation.

4. The method is validated only on BiomedCLIP and biomedical datasets. Although this focus is understandable, it limits claims about general applicability to broader VLM adaptation tasks. A small-scale validation on a general-domain model (e.g., OpenCLIP) would make the conclusions stronger.

5.  The ablation is mostly quantitative. A more interpretive discussion, e.g., how κ values vary across classes or what they reveal about semantic diversity. It would help connect the method back to its motivation.

6. Although the paper provides strong quantitative results, it offers little interpretive analysis of why the vMF-based alignment helps.
For instance, there is no examination of the learned κ (concentration) parameters, which could indicate how semantic compactness varies across classes or datasets. Without such insights, it is difficult to understand whether the improvements arise from genuinely better modeling of semantic dispersion or simply from an additional regularization effect.

7. The method relies on GPT-4 to generate 50 prompts per class. This introduces variability, cost, and potential bias that are not analyzed.
The paper does not clarify whether performance is sensitive to the quality, diversity, or domain appropriateness of the generated prompts, or how results would change if fewer or noisier prompts were used. Given the growing concern about reproducibility in LLM-assisted pipelines, this dependence deserves more systematic evaluation.

**Questions:**

See weaknesses.

---

### Official Review · Reviewer_34hv · 2025-10-31

**Soundness:** 2
**Presentation:** 2
**Contribution:** 2
**Rating:** 2
**Confidence:** 3

**Summary:**

The paper proposes vMF Distribution Semantic Alignment (VDSA) for adapting biomedical vision–language models (VLMs). Instead of collapsing multiple text prompts into a single prototype, each class’s text features are modeled with a von Mises–Fisher (vMF) distribution on the unit hypersphere, and image embeddings are aligned to this distribution via a sampling-free Jensen upper-bound objective. Experiments on 10 biomedical datasets claim consistent gains for few-shot learning and base-to-novel generalization over PEFT/prompting baselines (e.g., CoOp/CoCoOp, BiomedCoOp) using a BiomedCLIP ViT-B/16 backbone with LoRA tuning.

**Strengths:**

1. Clear motivation. The paper correctly identifies prototype averaging as lossy for prompt ensembles and motivates distributional modeling on the hypersphere.
2. Broad biomedical suite. The evaluation spans diverse modalities (MRI, CT, ultrasound, fundus, histology, X-ray), which is appropriate for the target domain.
3. The paper includes a basic ablation (VDSA vs. cross-entropy prototype alignment) showing consistent improvements.

**Weaknesses:**

Major Weakness:
1. Novelty. The paper positions prototype averaging as the core limitation and proposes a distributional alternative. However, recent works already emphasize richer prompt sets, LLM-generated ensembles, and knowledge-guided/gradient-guided prompts. The manuscript does not clearly differentiate VDSA from strong prompt-ensemble weighting or mixture-of-prompts approaches beyond the choice of vMF and a Jensen bound. Please clarify the concrete conceptual/algorithmic advances beyond “fit a vMF and replace the prototype” (e.g., why vMF over other spherical mixtures; why not learn class-conditional mixture weights).
2. Missing References. von Misers-Fisher loss has been studied in [1], Equation 8 in [1] and Equation 13 in [1]'s supplement, the approximated variate of the proposed $L_{VSDA}$ is used as the objective function.
3. Experiments. The ablation compares only VDSA vs. CE. Missing ablations include: effect of LoRA rank; learning $k_c$ vs. freezing MLE; comparing vMF to other directional families (e.g., Kent/Fisher–Bingham); learned vs. fixed $N$ prompts.

Minor Weakness:
1. Code/log (including the exact prompt lists, seeds, and vMF estimation routines) are not provided at review time. Given the sensitivity to prompt design and tuning, an anonymized repository with scripts to reproduce all tables/figures is essential.

[1] von Mises–Fisher Loss: An Exploration of Embedding Geometries for Supervised Learning. ICCV2021.

**Questions:**

1. Please clarify the concrete conceptual/algorithmic advances beyond “fit a vMF and replace the prototype”.
2. Could you please try to use the Equation 8 in [1] as the loss in your experiments?
3. Add more ablations.

---

> ### Author Response · Authors · 2025-11-20
>
> We thank the reviewer for the comments.  VDSA introduces a *distribution aware semantic alignment objective* that is fundamentally different from prompt ensemble weighting or mixture of prompts approaches. We highlight the concrete conceptual and algorithmic advances.
>
> **VDSA performs implicit infinite prompt augmentation, whereas previous methods operate only on a finite collection of prompts.**
>
> Existing prompt ensemble approaches aggregate **finite prompt embeddings** through linear weighting or discrete mixtures. These methods cannot capture the *continuous semantic variability* inherent in the prompt embedding space, their representation capacity is fundamentally bounded by the number of available prompts.
>
> In contrast, VDSA models prompts as a **continuous spherical density** using a vMF distribution. This induces an **implicit infinite augmentation effect**, meaning VDSA aligns image features not only to the given prompt embeddings but to the *entire continuous family* of plausible prompts characterized by the vMF density. Thus, VDSA is not a reweighting scheme; it shifts the paradigm from *finite sample aggregation* to *distribution level modeling*.
>
> This gives VDSA:
> - one interpretable semantic direction μ\muμ per class,
> - a stable Jensen upper bound yielding a *differentiable surrogate alignment loss*,
> - strict consistency with cosine based classification.
>
>
> **In summary**, VDSA differs from prior prompt-ensemble or mixture approaches by providing:
>
> (i) *implicit infinite prompt augmentation* through continuous density modeling, and
>
> (ii) a *geometry-aligned probabilistic objective* uniquely supported by the vMF distribution.

---

### Note · Authors · 2025-11-20

I have read and agree with the venue's withdrawal policy on behalf of myself and my co-authors.